# Urinary Biomarkers: Diagnostic Tools for Monitoring Athletes’ Health Status

**DOI:** 10.3390/ijerph17176065

**Published:** 2020-08-20

**Authors:** Raffaela Pero, Mariarita Brancaccio, Cristina Mennitti, Luca Gentile, Sergio Arpino, Renato De Falco, Eleonora Leggiero, Annaluisa Ranieri, Chiara Pagliuca, Roberta Colicchio, Paola Salvatore, Giovanni D’Alicandro, Giulia Frisso, Barbara Lombardo, Cristina Mazzaccara, Raffaella Faraonio, Olga Scudiero

**Affiliations:** 1Department of Molecular Medicine and Medical Biotechnology, University of Naples Federico II, Via S. Pansini 5, 80131 Naples, Italy; cristinamennitti@libero.it (C.M.); sergio.arp@libero.it (S.A.); renatodefalco@fastwebnet.it (R.D.F.); chiara.pagliuca@unina.it (C.P.); roberta.colicchio@unina.it (R.C.); paola.salvatore@unina.it (P.S.); giulia.frisso@unina.it (G.F.); barbara.lombardo@unina.it (B.L.); raffaella.faraonio@unina.it (R.F.); 2Task Force on Microbiome Studies, University of Naples Federico II, 80100 Naples, Italy; 3Department of Biology and Evolution of Marine Organisms, Stazione Zoologica Anton Dohrn, Villa Comunale, 80121 Naples, Italy; mariarita.brancaccio@szn.it; 4Ceinge Biotecnologie Avanzate S. C. a R. L., 80131 Naples, Italy; gentilelu@ceinge.unina.it (L.G.); leggiero@ceinge.unina.it (E.L.); anna.ranieri90@gmail.com (A.R.); 5Department of Neuroscience and Rehabilitation, Center of Sports Medicine and Disability, AORN, Santobono-Pausillipon, 80122 Naples, Italy; ninodalicandro@libero.it

**Keywords:** urine, biomarkers, élite athletes, infections, athlete’s health

## Abstract

Acute or intense exercise is sometimes related to infections of the urinary tract. It can also lead to incorrect hydration as well as incorrect glomerular filtration due to the presence of high-molecular-weight proteins that cause damage to the kidneys. In this context, our study lays the foundations for the use of a urine test in a team of twelve male basketball players as a means of monitoring numerous biochemical parameters, including pH, specific weight, color, appearance, presence of bacterial cells, presence of squamous cells, leukocytes, erythrocytes, proteins, glucose, ketones, bilirubin, hemoglobin, nitrite, and leukocyte esterase, to prevent and/or treat the onset of pathologies, prescribe personalized treatments for each athlete, and monitor the athletes’ health status.

## 1. Introduction

The analysis of stress induced by intense physical exercise encompasses the quantification and measurement of the parameters of strength, speed, power, and physical recovery to allow the coach and the medical health staff to calibrate the right training program for each athlete [1]. Recently, the scientific community has considered it appropriate to study biomarkers [2,3,4,5,6,7] capable of preventing muscle injuries [8,9,10], infections [11,12,13], and cardiovascular or thrombotic pathologies [14,15] while, at the same time, highlighting hidden genetic pathologies [16,17,18] that could undermine the athlete’s health. Another fundamental aspect of athlete care is the microbiome [19]. Recent studies have shown that safeguarding the microbiome in addition to the intestine and the immune system, including all its cellular and peptide components, can help to combat the occurrence of infections [20,21,22,23,24,25,26,27,28,29,30].

In this scenario, urine has been catalogued as containing valid and rapid biomarkers of pathologies that can affect the athlete by undermining physical integrity [31]. Urine possesses cellular and biochemical components that derive from plasma glomerular filtration, renal tubular excretion, and urogenital secretions [32]. Urinary proteins have been used as non-invasive biomarkers that can be used to accurately monitor body stress based on various psychophysiological changes and identify body conditions in intense, unusual physical exercise, competitions, excessive training, and improper recovery in sports [33,34]. Moreover, urine is an ideal source of biomarkers because it contains fewer lipids and a larger number of polypeptides than can be found in serum or tissues [35].

Furthermore, hydration levels affect urinary biomarker concentrations. Therefore, it is possible to highlight both conditions of hypohydration and hyperhydration in the athlete. Hypohydration conditions are known to affect and decrease plasma volume, cardiac activity, sweating, cutaneous blood flow, and stamina. Conversely, hyperhydration conditions can cause brain nausea (a sensation of an urge to vomit that is controlled by the brain) and confusion due to the enlargement of the brain via an overdose of water. This is why monitoring an athlete’s hydration is essential [36,37]. Moreover, intense exercise causes the expenditure of electrolytes and carbohydrates and affects the body’s temperature [35,36,37,38,39,40].

In recent years, the scientific community has become aware that intense and prolonged exercise can lead to opportunistic infections [41,42,43]. It is known that urinary tract infections (UTIs) can occur in the presence of pathogenic microorganisms in the urine in an amount that can damage the urinary tract and, subsequently, if the infection is prolonged, the whole organism, with peculiar consequent symptoms and signs.

Urinary tract infections are one of the most common types of infections. Women are more affected [44] than men, and, for women, the risk of contracting a urinary tract infection throughout life is 50%. Indeed, one-third of all women will be affected by a UTI before the age of twenty-four. Moreover, many women have repeated infections, sometimes over a period of years.

Recent studies have identified a resident microbial community in the urinary tract (UT). The contribution of the UT microbiome to a UTI is not yet clearly understood. It has been suggested that commensal species within the UT microbiome and the urogenital tract (UGT) microbiome, such as *Lactobacillus crispatus*, may play an important role in protection against colonization by uropathogens [45].

It is also crucial to recognize and treat urinary tract infections in order to prevent them from becoming complicated by infecting essential organs of the body. However, above all, it is essential to treat them correctly, as resistant bacteria can withstand treatment.

Bacterial resistance is one of the most dangerous clinical challenges. Therefore, rapid identification of the pathogen is necessary for the provision of adequate care. Athletes who share toilets, showers, and changing rooms can more easily contract these infections [46].

In some cases, pathogenic microorganisms enter the lower urinary tract (the bladder) through the venous and lymphatic pathways [47]. Bacteria can also invade and colonize the bladder through the urethra [48,49]. However, the main route used by bacteria that come from the large intestine is the urethra (the channel that carries urine from the bladder to the outside of the body). The most common of these bacteria is *Escherichia coli*, which is found in 75% of UTIs in men and 95% of UTIs in women. Less common pathogens include other *Enterobacteriaceae*, such as *Klebsiella pneumoniae* and *Proteus mirabilis* [50]. Bacteria such as *Pseudomonas*, *Enterococci*, and *Staphylococci* have also been found in athletes with recent hospitalizations [51]. Generally, this type of infection affects the lower urinary tract, i.e., the urethra and bladder. However, if these infections are not adequately treated, they can spread to the upper urinary system or to the ureters (the ducts that transport urine from the kidneys to the bladder) and the kidneys. This can lead to a bladder infection, called cystitis, which is the most common type of urinary tract infection; a urethra infection, also known as urethritis; a ureter infection, i.e., urethritis; or a kidney infection (pyelonephritis), which is a severe pathology that requires an immediate intervention. If left untreated, pyelonephritis can lead to a loss of kidney function and, in severe cases, even to death. [52].

Unfortunately, 20% of all people, but especially women, experience a second infection after the first (a recurrent infection), and some people suffer for a long time and incessantly (recurrent infections) [53]. Sometimes, the bacterial strains in cases of recurrence are different from those that caused the first infection (acute infection) [54]. On the other hand, uropathogens invade the structures of the urinary tract, where they may become chronically present and find themselves protected from drug therapies. Thus, screening of the urine of athletes can help to identify the infection and the bacterium and provide an immediate targeted antibiotic treatment. UTIs are classified as complicated and non-complicated. When a complicated UTI is suspected or the pathogen is resistant to first-choice therapy agents, the antibiotics commonly used to treat the UTI are those belonging to the class of fluoroquinolones (nalidixic acid, norfloxacin, ciprofloxacin, levofloxacin, and ofloxacin) [55,56].

Exercise can also induce kidney dysfunction, causing a series of changes that include proteinuria, hematuria, a reduced renal blood flow, and a reduced glomerular filtration rate. Several issues can arise due to functional or temporary proteinuria after physical activity [50]. In athletes, a higher incidence of proteinuria has been observed to be caused by intense and prolonged exercise and is linked to the intensity of muscle work. Exercise-induced proteinuria is more closely related to exercise intensity than to exercise duration [57,58,59,60,61]. Strenuous exercise can be responsible for a severe form of skeletal muscle damage known as Exertional Rhabdomyolysis (ER), which is characterized by the breakdown of muscle cells and the release of the substances contained in the circulation. Although ER is a relatively rare condition, it can lead to the severe complications of muscle ischemia, cardiac arrhythmia, and death [62]. The onset of ER is related to the intensity, duration, and type of exercise, and to other essential factors such as temperature and humidity during exercise. For the diagnosis of ER, creatine kinase (CK) levels represent the most sensitive marker, and excessive myoglobin concentrations contained in the urine may exhibit myoglobinuria with dark colors [63]. As regards hematuria, its incidence is higher in athletes than in the general population. The main distinction is that sports-related hematuria settles spontaneously after exercise while hematuria discovered in a non-athletic person can be chronic [64,65]. Sports-induced hematuria is affected by the duration and intensity of the exercise. The mechanisms underlying exercise-induced hematuria include increased body temperature, hemolysis, increased production of free radicals, and excessive release of catecholamines [66,67,68]. Another fundamental factor is lactic acidosis, which is generated under anaerobic conditions. It causes the passage of erythrocytes in the urine through a higher glomerular permeability [68].

This study aimed to examine the urinary modifications in basketball players during training, competition, and the recovery period in order to verify if urine could be used to develop a valid, reliable, and sensitive sport-specific test for the continuous monitoring of athletes.

## 2. Materials and Methods

### 2.1. Experimental Approach

We evaluated via urine the possible infections affecting professional basketball players using the stress caused by physical exercise during a season of competition as a dependent variable. The study was designed following the recommendations for clinical research contained in the Helsinki Declaration of the World Medical Association. The protocol was approved by the Ethics Committee of the School of Medicine, University of Naples Federico II (protocol number 200/17).

### 2.2. Participants

We included professional basketball players (*n* = 12) in this study. Participants were men and were informed about the research and the protocols used. The physical characteristics of the players, expressed as the mean (±SD), were: age, 25 ± 6 (years); weight, 92 ± 10 (kg); and height, 195 ± 9 (cm). None of the subjects smoked, drank alcohol, or took drugs known to alter the chemical parameters of the leukocyte or hormonal formula. All individuals followed a similar diet throughout the season and, most importantly, the same diet during the study period. The participants’ diet was monitored continuously by the team doctors. The players followed the same training program. They trained every day in two sessions: a morning session that consisted of a 2-h gym workout and an afternoon session that consisted of a 3-h basketball practice period. This training program was followed daily, except for days on which official games were played during the season (two games per week).

### 2.3. Urine Sampling

Urine samples from professional athletes were taken at 0 months (i.e., during the preseason phase), one month after the start of the season, and three months after the start of the season. We evaluated the following parameters of the urine samples: pH (5.5–7 mg/dL), specific weight (1005–1030), color, appearance, presence of bacterial cells (0–1000 n/µL), presence of squamous cells (0–20 n/µL), leukocytes (0–18 n/µL), erythrocytes (0–14 n/µL), proteins (0–20 mg/dL), glucose, ketones, bilirubin, hemoglobin, nitrite, and leukocyte esterase. The urine samples were taken in the morning before training for all athletes. Before proceeding with the collection of urine, we recommended that each athlete: (1) wash their hands thoroughly with soap and water and dry them with a cloth; (2) wash their genital organs thoroughly with soap and water (after having retracted the foreskin on the glans) and dry them; and (3) discard the first jet of urine in the W.C. and collect the second jet of urine in a sterile container without interrupting the urination.

### 2.4. Biochemical Determinations

First-morning urine samples were collected from athletes and analyzed within 4 h of arrival by the combined use of an automated urine chemistry analyzer (UC3500, Sysmex, Kobe, Japan) and a fluorescence flow cytometer (UF 1000i, Sysmex, Kobe, Japan) according to the manufacturer’s instructions. Where necessary, we conducted an examination using an optical microscope.

### 2.5. Statistical Analysis

All statistical analyses were performed using GraphPad Prism 8.4.0 software (GraphPad Software Inc., La Jolla, CA, USA). Data are expressed as the mean ± standard deviation. As appropriate, comparisons among groups were made by analysis of variance ANOVA followed by Bonferroni or Tukey’s multiple comparison tests. Values of *p* < 0.05 were considered significant.

## 3. Results

### 3.1. Physical Activity Modifies Hydration Status in Professional Athletes

To evaluate the hydration status of athletes, we analyzed the urinary parameters pH (Figure 1A and Table 1), specific weight (Figure 1B and Table 1), color (Table 1), and appearance (Table 1) over a three-month period in a group of 12 professional basketball players. At 0 months, 50% of the athletes had a pH ≥ 5; 33% had a pH ≥ 6, and 17% had a pH ≥ 7 (Table 1). At one month, the pH values were found to have undergone a change. In fact, 31% of the athletes had a pH ≥ 5; 38% had a pH ≥ 6, and 31% had a pH ≥ 7 (Table 1). At three months, 59% of the athletes had a pH ≥ 5; 8% had a pH ≥ 6, and 33% had a pH ≥ 7 (Table 1). Furthermore, we assessed the specific weight (SW). At 0 months, 10% of the athletes had an SW ≥ 1005; 50% had an SW ≥ 1018, and 40% had an SW ≥ 1030 (Table 1). At one month, the SW values showed changes. In fact, 10% of the athletes had an SW ≥ 1005, 60% had an SW ≥ 1018, and 30% had an SW ≥ 1030 (Table 1). Finally, at three months, 50% of the athletes had an SW ≥ 1018, and the remaining 50% had an SW ≥ 1030 (Table 1). In addition, we evaluated the color. At 0 months, 92% of the athletes had urine with a straw yellow color and the remaining 8% had urine with an amber color (Table 1). At one month, 83% of the athletes had urine with a straw yellow color and 17% had urine with an amber color (Table 1). At three months, 100% of the population considered had urine with a straw yellow color (Table 1). Moreover, we evaluated appearance. At 0 months, 83% of the athletes had urine with a clear appearance, 9% had urine with an opalescent appearance, and 8% had urine with a cloudy appearance (Table 1). At one month, 75% of the athletes had urine with a clear appearance, 8% had urine with an opalescent appearance, and 17% had urine with a cloudy appearance (Table 1). By the third month, all of the athletes examined (100%) had urine with a clear appearance (Table 1). Finally, we carried out a statistical analysis of the values of pH and specific weight (Figure 1A,B). In this case, the urine pH of the 12 competitive athletes decreased significantly with increasing time (Figure 1A), whereas the specific weight increased significantly with increasing time.

### 3.2. Physical Activity Modifies Susceptibility to Urinary Infections

To monitor the presence or absence of bacterial populations within the group of professional athletes, we considered the following urinary parameters: bacteria (B) (Figure 2A and Table 2), squamous cells (SC) (Figure 2B and Table 2), and leukocytes (L) (Figure 2C and Table 2).

At 0 months, 83% of the athletes had a bacterial population between 0 and 200 n/µL, 9% had a bacterial population between 201 and 500 n/µL, and 8% had a bacterial population between 501 and 1000 n/µL (Table 2). At one month, 67% of the athletes had a bacterial population between 0 and 200 n/µL, 8% had a bacterial population between 201 and 500 n/µL, and 25% had a bacterial population between 501 and 1000 n/µL (Table 2). At three months, 100% of the athletes had a bacterial population between 0 and 200 n/µL (Table 2).

Moreover, we evaluated the SC. At 0 months, 75% of the athletes had an SC population between 0 and 5 n/μL, 8% had an SC population between 6 and 10 n/µL, and 17% had an SC population between 11 and 20 n/μL (Table 2). At one month, 50% of the athletes showed an SC population between 0 and 6 n/µL, 25% showed an SC population between 6 and 10 n/µL, and the remaining 25% showed an SC population between 11 and 20 n/µL (Table 2). At three months, however, 75% of the athletes had an SC population between 0 and 5 n/μL, 17% had a SC population between 6 and 10 n/µL, and 8% had an SC population between 11 and 20 n/μL (Table 2).

In addition, at 0 months, 58% of the athletes had a leukocyte population between 0 and 4 n/μL, 17% had a leukocyte population between 5 and 9 n/µL, and 25% had a leukocyte population between 10 and 18 n/µL (Table 2). At one month, 50% of the athletes had a white blood cell population between 0 and 4 n/µL, 8% had a white blood cell population between 5 and 9 n/μL, and 42% had a white blood cell population between 10 and 18 n/µL (Table 2). At three months, 67% of the athletes had a leukocyte population between 0 and 4 n/μL, 8% had a leukocyte population between 5 and 9 n/µL, and 25% had a leukocyte population between 10 and 18 n/µL (Table 2).

Furthermore, on the 12 urine samples obtained from the athletes, gram staining was performed; thus through optical microscopy we identified the presence of gram negative bacteria. Later, to determine the type of bacterium, we performed culture examination of the sample colonies on McConkey agar to observe whether or not lactose fermentation had taken place. In our case, the test was positive; we then continued with the methyl red test (MR-VP) to verify if the highlighted bacteria could carry out glucose fermentation, and this too was positive. Finally, through the electron microscope we carried out analysis of the colonies, highlighting the presence of *Escherichia coli* in all samples containing bacteria [69,70].

Finally, we carried out a statistical analysis of the values for bacteria (B), squamous cells (SC), and leukocytes (L) (Figure 2A–C). In this case, the three analyzed parameters showed the same trend (Figure 2A–C). In fact, the number of bacteria, the number of squamous cells, and the number of leukocytes all decreased significantly with increasing time (Figure 2A–C respectively).

### 3.3. Physical Activity Modifies Hematuria and Proteinuria in Professional Athletes

To shed light on the presence of hematuria and/or proteinuria, we evaluated the following urinary parameters: erythrocytes (E) (Figure 3A and Table 3) and proteins (P) (Figure 3B and Table 3).

At 0 months, 50% of the athletes had an erythrocyte population (E) between 0 and 3 n/μL, 33% had an E between 4 and 7 n/μL, and 17% had an E between 8 and 14 n/μL (Table 3). At one month, 50% of the athletes had an E between 0 and 3 n/μL, 25% had an E between 4 and 7 n/μL, and the remaining 25% had an E between 8 and 14 n/μL (Table 3). At three months, 75% of the athletes had an erythrocyte population (E) between 0 and 3 n/μL, 17% had an E between 4 and 7 n/μL, and 8% had an E between 8 and 14 n/μL (Table 3).

Furthermore, we evaluated the protein (P) content. At 0 months, 75% of the athletes had P content between 0 and 15 mg/dL, 17% P content between 16 and 50 mg/dL, and 8% P content between 51 and 100 mg/dL (Table 3). At one month, 75% of the athletes showed P content between 0 and 15 mg/dL, 8% showed P content between 16 and 50 mg/dL, and 17% showed P content between 51 and 100 mg/dL (Table 3). At three months, 75% of the athletes had a P value between 0 and 15 mg/dL and the remaining 25% had a P value between 16 and 50 mg/dL (Table 3).

Finally, we carried out a statistical analysis of the values for erythrocytes (E) and proteins (P) (Figure 3A,B). In this case, the numbers of both erythrocytes and proteins decreased with increasing time.

### 3.4. Physical Activity Does Not Affect Urinary Parameters in Professional Athletes

To evaluate the general health of the athletes, we also monitored the following parameters: glucose, ketones, bilirubin, hemoglobin, nitrite, and leukocyte esterase.

In the three months of monitoring, none of the parameters were detected within the examined population of professional athletes (Table 4).

## 4. Discussion

Laboratory medicine plays a central role in sport and monitoring athletes’ health [1,66,67]. Numerous scientific studies have shown that intense and prolonged exercise can cause metabolic adaptations. The variations in these parameters represent the changes that occur in the body in response to the intensity and duration of physical exercise as well as the stress to which the different organs and muscle units involved in athletic performance are subjected. These adaptations translate into alterations in specific parameters, in terms of concentration and activity, and their identification could represent a new method for monitoring the health of an athlete. We have attempted to shed light on changes in urinary biochemical parameters and evaluate different parameters since the response to stress induced by exercise implies the involvement of organs and tissues. To the best of our knowledge, no study in the literature has ever integrated all of these different aspects. To make our observations, we took urine samples from a group of professional male basketball athletes.

Firstly, we assessed each athlete’s hydration status, analyzing pH, specific weight, color, and appearance. It is known that insufficient hydration (hypohydration) or excessive hydration (hyperhydration) can be harmful to an athlete’s performance [35,36,37,38,39,40]. Our data showed that at 0 months (i.e., during the training period), 83% of the athletes had 5 ≤ pH ≤ 6 or a normal pH and that only 17% had a pH ≥ 7. At the end of the athletic preparation period and at the start of the season, the situation was found to be slightly different; in fact, 69% of the athletes had 5 ≤ pH ≤ 6, while 31% had a pH ≥ 7, and, at three months, 66% had 5 ≤ pH ≤ 6 and 33% had a pH ≥7. It is essential to note that the percentage of athletes with a pH ≥ 7 increased by 14% between month 0 and month 1, and by 16% between month 0 and month three; however, between month one and month three, there was an increase of only 2%. When an individual shows a pH value greater than 7, he presents an alkaline pH, which often coincides with the presence of a urinary tract infection. The inverse trend is shown in the population of athletes with 5 ≤ pH ≤ 6. So, the number of cases of diarrhea or starvation will undoubtedly be reduced because a pH lower than 5 can cause this pathological manifestation. Furthermore, the statistical analysis carried out showed that the pH decreased; therefore, with increasing physical activity, it is necessary to monitor the hydration of athletes in order to avoid the condition of pathological hypohydration. As regards the specific weight, 60% of the athletes at 0 months showed values 1005 ≤ SW≤ 1018, while only 40% had an SW ≥ 1030; at one month, 70% had 1005 ≤ SW ≤ 1018 and 30% had an SW ≥ 1030; finally, at three months, 50% had an SW ≥ 1018 and 50% had an SW ≥ 1030. Therefore, it is clear that there was a decrease in the specific weight over time. This could have been due to the slight dehydration caused by the intensity of the physical exercise.

This trend was also confirmed by the statistical analysis carried out. We then evaluated the color of the urine. At 0 months, 92% of the athletes had urine with a straw yellow color and 8% had urine with an amber color. At one month, the population with straw yellow urine decreased by 9% and, consequently, the population with amber urine increased. However, at three months, the entire population of athletes examined (100%) had urine with a straw yellow color, which, therefore, reflected that the urinary tract was in an excellent state of health. Moreover, we evaluated the appearance of the urine. At 0 months, 83% of the athletes had urine with a bright appearance, 9% had urine with an opalescent appearance, and 8% had urine with a cloudy appearance. At one month, the population with clear urine decreased by 8% and that with cloudy urine increased. By three months, the whole population had urine with a bright appearance. The analysis of these four parameters suggested that, between month 0 and month one, the athletes slightly lacked hydration, most likely because their bodies needed to get used to the intensity of the training. At the same time, sharing spaces can lead to the occurrence of some urinary tract infections that alkalize the pH, increase the specific weight, and cloud the color and appearance of urine; a condition that resolves after three months when the body has adapted to training stress and possible infections have been eradicated with appropriate treatments.

To evaluate the susceptibility to a UTI [41,42,43,44,48,49,51,55], we evaluated three parameters: the presence of bacteria; the presence of squamous cells; and the presence of leukocytes. It is known that in cases of urinary tract infection there is an increase in the number of bacteria and, consequently, an increase in the white blood cell population because the immune system is activated. In some cases, squamous cells are present. Squamous cells, in numbers outside the norm, deserve particular attention as they could be a prelude to a more severe pathological condition ranging from a common infection to cancer. In our case, at 0 months, only 8% of the athletes showed a bacterial population between 501 and 1000 n/µL, i.e., a pathological population, which increased to a value of 25% at one month. In contrast, at three months, 100% of the examined population had values between 0 and 200 n/µL, i.e., normal values. The leukocyte population followed the same trend. In fact, at 0 months, 25% of the athletes had values between 10 and 18 n/μL, which are borderline normal values; at one month, the leukocyte population increased by 17% and then decreased by the same percentage in the third month. The same applies to squamous cells, which had values between 11 and 20 n/μL (17% (0 months), 25% (one month), and 8% (three months)). Therefore, while between month 0 and month one we observed an increase of 8%, between month 0 and month three we observed a decrease of 9%, which became even more evident between month one and month three (a value equal to 17%). These three parameters reinforce the idea that athletes need a period of physical and immune adaptation and that this period of adaptation lasts for about three months. In fact, in the statistical analysis carried out, the three parameters showed the same trend or an increase to month one and then a decrease to month three.

It is known that competitive activity can cause hematuria and proteinuria [50,57,58,59,60,61,64], which is why we evaluated both erythrocytes and proteins in the urine. Our data show that at 0 months, 17% of the athletes had a slightly borderline number of erythrocytes in the urine, ranging from 8 to 14 n/μL; at 1 month, this population increased by 8%; and, finally, at three months, there was a reduction of 17%. At the same time, we found a population equal to 25% of proteins in the third month that had borderline values between 16 and 50 mg/dL, which is an increase compared with month 0 (by 17%) and compared with month one (by 8%). However, at three months, 75% of the athletes had values in the normal 0–15 mg/dL range. These data suggest that hematuria and proteinuria decrease over time, as shown in the statistical analysis.

These values do not indicate glomerular damage [65,66,67,68,71], as we evaluated six other parameters: glucose, ketones, bilirubin, hemoglobin, nitrite, and leukocyte esterase. We found these parameters were always absent over the three-month study period.

Therefore, we can say that the urine test allowed us to monitor the health of the athletes, highlighting that between 0 and one months an athlete’s body and, therefore, the organs that constitute it, undergo stimuli that can lead to fatigue, reduced hydration, the appearance of infections, and slight hematuria and proteinuria. Then, these parameters seem to fall away entirely after three months.

## 5. Conclusions

The biochemical parameters analyzed in this work were used as indices to evaluate and monitor the physical condition of professional athletes over three months, to protect the athletes’ health, and also to prevent the athletes’ performance from deteriorating due to the onset of urinary tract disease. This work lays the foundation for outlining how a simple and inexpensive laboratory investigation such as urinalysis can generate a panel of specific markers that can be applied to individual athletes. The results obtained would guarantee for each athlete the prevention of new infections and timely targeted antibiotic treatment by sports doctors of existing infections, thereby reducing the continuation of infections that can be aggravated by intense physical exercise.

In conclusion, the urine test could be included in the laboratory analyzes [72] that would safeguard the athlete’s health and improve physical performance.

## Figures and Tables

**Figure 1 ijerph-17-06065-f001:**
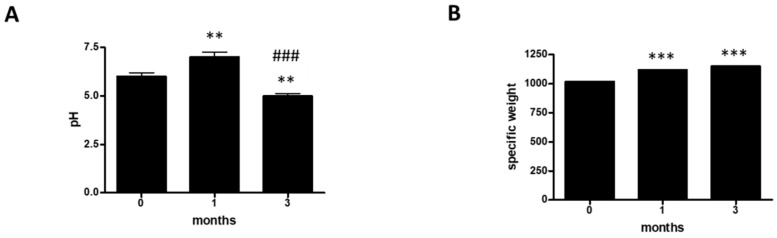
Hydration status. (**A**) Assessment of urine pH at 0, 1, and 3 months in a group of 12 professional basketball athletes. The data are expressed as the mean ± SD. The significance was determined by the one-way ANOVA test: ** (*p* < 0.01) represents significance compared with 0 months; ^###^ (*p* < 0.001) represents significance compared with 1 month. (**B**) assessment of specific weight at 0, 1, and 3 months in a group of 12 professional basketball athletes. The data are expressed as the mean ±SD. The significance was determined by the one-way ANOVA test *** (*p* < 0.001) represents significance compared with 0 months.

**Figure 2 ijerph-17-06065-f002:**
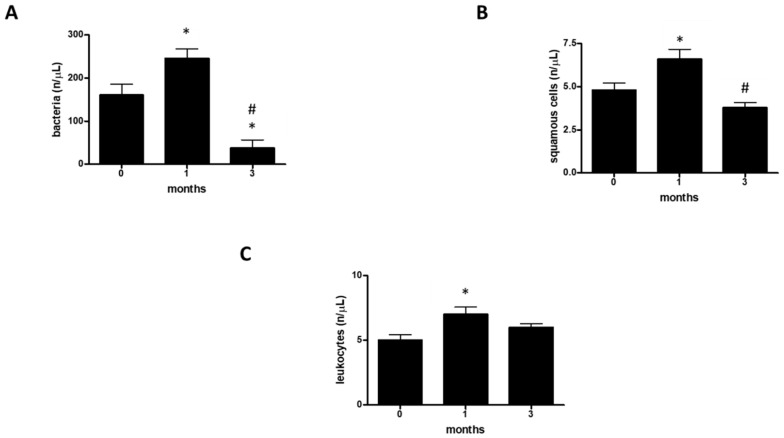
Urinary infections. (**A**) assessment of bacteria at 0, 1, and 3 months in a group of 12 professional basketball athletes. The data are expressed as the mean ±SD. The significance was determined by the one-way ANOVA test: * (*p* < 0.05) represents significance compared with 0 months; ^#^ (*p* < 0.05) represents significance compared with 1 month. (**B**) assessment of squamous cells at 0, 1, and 3 months in a group of 12 professional basketball athletes. The data are expressed as the mean ± SD. The significance was determined by the one-way ANOVA test: * (*p* < 0.05) represents significance compared with 0 months; ^#^ (*p* < 0.05) represents significance compared with 1 month. (**C**) assessment of leukocytes at 0, 1, and 3 months in a group of 12 professional basketball athletes. The data are expressed as the mean ± SD. The significance was determined by the one-way ANOVA test: * (*p* < 0.05) represents significance compared with 0 months. The unit of measurement used is n/μL (the number of bacteria or leukocytes or squamous cells in one microliter of urine).

**Figure 3 ijerph-17-06065-f003:**
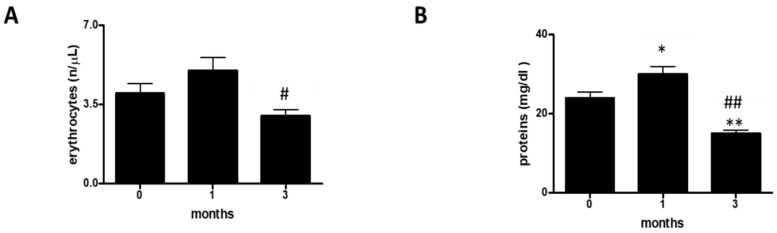
Hematuria and proteinuria. (**A**) assessment of erythrocytes at 0, 1, and 3 months in a group of 12 professional basketball athletes. The data are expressed as the mean ±SD. The significance was determined by the one-way ANOVA test: ^#^ (*p* < 0.05) represents significance compared with one month. (**B**) assessment of proteins at 0, 1, and 3 months in a group of 12 professional basketball athletes. The significance was determined by the one-way ANOVA test: * (*p* < 0.05) and ** (*p* < 0.01) represent significance compared with 0 months; ^##^ (*p* < 0.01) represents significance compared with one month. The unit of measurement used is n/μL (the number of bacteria or leukocytes or squamous cells in one microliter of urine).

**Table 1 ijerph-17-06065-t001:** Hydration parameters. Percentage representation of pH, specific weight, color and appearance of urine in a group of 12 professional athletes.

Parameters	Values	0 Months	1 Months	3 Months
pH	≥5	50%	31%	59%
	≥6	33%	38%	8%
	≥7	17%	31%	33%
Specific Weight	≥1005	10%	10%	
	≥1018	50%	60%	50%
	≥1030	40%	30%	50%
Color	Amber	8%	17%	
	Straw yellow	92%	83%	100%
Appearance	Clear	83%	75%	100%
	Opalescent	9%	8%	
	Turbid	8%	17%	

**Table 2 ijerph-17-06065-t002:** Urine infection parameters. Percentage of bacterial populations, presence of erythrocytes and leukocytes in a group of 12 professional athletes.

Parameters	Values	0 Months	1 Months	3 Months
Bacteria	0–200 n/µL	83%	67%	100%
	201–500 n/µL	9%	8%	
	501–1000 n/µl	8%	25%	
Squamous Cells	0–5 n/μL	75%	50%	75%
	6–10 n/µL	8%	25%	17%
	11–20 n/μL	17%	25%	8%
Leucocytes	0–4 n/μL	58%	50%	67%
	5–9 n/µL	17%	8%	8%
	10–18 n/µL	25%	42%	25%

**Table 3 ijerph-17-06065-t003:** Hematuria and proteinuria parameters. Percentage of hematuria and proteinuria in a group of 12 professional athletes.

Parameters	Values	0 Months	1 Months	3 Months
Erythrocytes	0–3 n/μL	50%	50%	75%
	4–7 n/μL	33%	25%	17%
	8–14 n/μL	17%	25%	8%
Protein	0–15 mg/dL	75%	75%	75%
	16–50 mg/dL	17%	8%	25%
	51–100 mg/dL	8%	17%	

**Table 4 ijerph-17-06065-t004:** Evaluation of urinary parameters.

Variables	0 Months	1 Month	3 Months
Glucose	absent	absent	Absent
Ketones	negative	negative	Negative
Bilirubin	absent	absent	Absent
Hemoglobin	absent	absent	Absent
Nitrite	absent	absent	Absent
Leukocyte esterase	absent	absent	Absent

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
