# Peer review of "Urinary Biomarkers: Diagnostic Tools for Monitoring Athletes’ Health Status"

_ijerph, 2020, doi:10.3390/ijerph17176065_

Round 1

Reviewer 1 Report

All my comments have been addressed.  I have no further comments.

Author Response

All my comments have been addressed. I have no further comments.

We thank the Reviewer for the response.

Reviewer 2 Report

The paper of Pero et al. analyses the changes of the major clinical urinary markers during extensive and prolonged physical activity. The paper itself aims to correlate the biomarkers and establish the infection-specific markers to prognoses the infectious processes in urinary tract.

Although, the authors did not fully respond to the previous comments, the current manuscript was modified to partly address the major comments.

Below I am providing my major comments and suggestions regarding current manuscript.

L44 requires attention. Perhaps there were some formatting problems ( co μL d)

L48-49 Authors stating: In this scenario, the urine has been catalogued as valid and rapid biomarkers to highlight any pathologies that can affect the athlete by undermining his physical integrity [31].
Please clarify the sentence.

L50 requires attention due to strange symbols.

L59 please provide the explanation of the condition indicated as brain nausea.

L64 I would recommend to shorten the sentence by changing: to the appearance of opportunistic infectious diseases [41-43]. To opportunistic infections.

L66 The term pathogenic microorganisms already indicating that the microorganisms will be harmful. Therefore please consider removing the explanation in between ().

L66-L67 the sentence requires re-phrasing. One might argue the extent of the damage to the urinary tract and whole organism. The systemic manifestation (is a result of prolonged infection) is a multistep process.

L67-L69 please consider rephrasing by simply stating that urinary tract infections are one of the most common <..>.

L73-L75 ,, However, above all, it is essential to treat them 74 correctly: inadequate or incomplete therapy favours bacterial resistance; that is, it offers bacteria the 75 possibility to defend themselves becoming insensitive to antibiotics”

One might argue whether the bacteria is defending themselves. I would recommend to re=phrase the sentence by stating that resistant bacteria/pathogens/organism can withstand/survive the treatment <..>.

L76-L78 Plase provide the reference stating that in this particular population (athletes)  the prevalence of the infections/organisms is higher than in other population.

L79 perhaps in some cases?

L79 I do not agree with proposed mechanism of infection. PLEASE PROVIDE the paper or other suitable reference indicating such pathways as a major cause of the UTI. Up to my knowledge the statement is incorrect. Authors included two references suggested them for reading as a supportive information. The included references are NOT mentioning venous and lymphatic pathways what so ever. This must be fixed.

L82 there are regulations for microbial taxonomy writing. Please find them at the MDPI guidelines for authors. Please fix the name of the microorganism.

L84-L87 The taxonomy must be fixed. Please make sure that you are using the correct term of Enterobacteriacae or Enterobacteriales. The statement needs reference.

L87 the reference No 47 (Echols, R.M.; Tosiello, R.L.; Haverstock, D.C.; Tice, A.D. DemograpHic, clinical, and treatment parameters 493 influencing the outcome of acute cystitis. Clin Infect Dis 1999, 29(1), 113.) is not providing data supporting the statement (In athletes with recent 86 hospitalizations, bacterial such as Pseudomonas, Enterococci and Staphylococci have also 87 been found [47].)

Moreover the taxonomic writing of the organisms must be fixed.

L87-L93 the part requires attention and re-writing/clarification. The part is missing reference indicating the source of the information.

L93-95 Unfortunately, 20% of 94 people experience a second infection after the first (recurrent infections), and some people suffer from 95 it for a long time and incessantly (recurrent infections).

It is unclear where authors find the results. The reference must be included.

L95 Sometimes, the bacterial strains in cases of recurrence are different from those that have infected the first time (acute infection), on the other 97 hand, uropathogens invade the structures of the urinary tract, becoming a chronic presence, 98 therefore, they find themselves protected from drug therapies

Why sometimes the strain is different while other times the strain is the same? The reference is missing.

L98-L99 Therefore, a urine screening of athletes 99 can help identify the infection, the bacterium and for an immediate targeted antibiotic treatment.

Why in this case athletes are selected population with increased demand to be screened for recurrent UTI’s?

L105 requires attention.

The overall introduction requires great attention.

L140 requires attention.

2.3 Are those values in between the brackets indicating the diagnostic range or normal values?
L151 proteins (0 e 20 mg/dl) should e be replaced by -?

L162 authors did not provided the rationale when the study design required to use a microscopy.

L166 p should be italic.

Figure 1. Again, I would recommend to simplify the presentation of the data by changing the style of the graphs to column graphs. Di authors used Grappad to prepare such graphs as indicated by the methods part?

The units presented in the pie graphs are unknown. It is not clear what units were used for the graph construction. The description indicates that authors used mean ±SD, although I am not sure how mean values were calculated. Please provide the clear statement.

L203 authors did not monitor bacterial populations. Please fix the statement, since the used methodologies are not allowing to monitor the bacterial populations.

L206 the same problem regarding bacterial populations.

L220—L221 Furthermore, from a microscopic analysis, of the samples containing bacteria we have 221 highlighted the presence of Escherichia Coli.

The taxonomy is wrong. Moreover and the most important, the microscopy will NEVER allow one to identify the species. All that is possible (without staining) is to differentiate the morphology. The statement is very wrong and not correct in many aspects.

 Fig 2. Fig 3 again the units are not clear. Same problem as Fig 1.

Discussion

The manuscript lacks of discussion of the results. The authors do not fully compares their results with similar results observed by other authors.
It is still unclear why authors decided to analyze male subjects and not females or both gender.  There is no provided rationale for the subject selection in the discussion.

The manuscript contains many typos.

Author Response

Reviewer 2

Comments and Suggestions for Authors

The paper of Pero et al. analyses the changes of the major clinical urinary markers during extensive and prolonged physical activity. The paper itself aims to correlate the biomarkers and establish the infection-specific markers to prognoses the infectious processes in urinary tract.

Although, the authors did not fully respond to the previous comments, the current manuscript was modified to partly address the major comments.

Below I am providing my major comments and suggestions regarding current manuscript.

L44 requires attention. Perhaps there were some formatting problems ( co μL d)

Response:

Thanks to the Reviewer 2 for the helpful comments.

In order to satisfy our request, we have modified the text: current line 42.

L48-49 Authors stating: In this scenario, the urine has been catalogued as valid and rapid biomarkers to highlight any pathologies that can affect the athlete by undermining his physical integrity [31].

Please clarify the sentence.

Response:

Through the simple examination of the urine it is possible to highlight alterations with respect to normal conditions and to deepen, if necessary, with specialist tests in order to assess the general health of the athlete.

L50 requires attention due to strange symbols.

In order to satisfy our request, we have now modified the text: current line 49.

L59 please provide the explanation of the condition indicated as brain nausea.

Response:

In order to satisfy our request, we have modified the text current lines 58-59.

Brain nausea is a condition characterized by a sensation of an urge to vomit controlled by the brain.

L64 I would recommend to shorten the sentence by changing: to the appearance of opportunistic infectious diseases [41-43]. To opportunistic infections.

Response:

In order to satisfy your request, we have modified the text: current line 64.

L66 The term pathogenic microorganisms already indicating that the microorganisms will be harmful. Therefore, please consider removing the explanation in between ().

Response:

In order to satisfy you request, we have modified the text: current line 65.

L66-L67 the sentence requires re-phrasing. One might argue the extent of the damage to the urinary tract and whole organism. The systemic manifestation (is a result of prolonged infection) is a multistep process.

Response:

In order to satisfy you request, we have modified the text from current line 65 to 67.

L67-L69 please consider rephrasing by simply stating that urinary tract infections are one of the most common <..>.

Response:

In order to satisfy you request, we modified the text: current line 68 .

L73-L75 ,, However, above all, it is essential to treat them 74 correctly: inadequate or incomplete therapy favours bacterial resistance; that is, it offers bacteria the 75 possibility to defend themselves becoming insensitive to antibiotics”

One might argue whether the bacteria is defending themselves. I would recommend to re=phrase the sentence by stating that resistant bacteria/pathogens/organism can withstand/survive the treatment <..>.

Response:

In order to satisfy you request, we modified the text: current line 78.

L76-L78 Plase provide the reference stating that in this particular population (athletes) the prevalence of the infections/organisms is higher than in other population.

Response:

In order to satisfy you request, we have added the following reference:

Zahra Ahmadinejad; Neda Alijani; Sedigeh Mansori; Vahid Ziaee. Common Sports-Related Infections: A Review on Clinical Pictures, Management and Time to Return to Sports. Asian Journal of Sports Medicine, 2014, 5(1): 1-9.

L79 perhaps in some cases?

Response:

There was a transcriptional error, we meant in some case. For this we have corrected line 78.

L79 I do not agree with proposed mechanism of infection. PLEASE PROVIDE the paper or other suitable reference indicating such pathways as a major cause of the UTI. Up to my knowledge the statement is incorrect. Authors included two references suggested them for reading as a supportive information. The included references are NOT mentioning venous and lymphatic pathways what so ever. This must be fixed.

Response:

In order to satisfy your request, we have added the following reference:

Wagenlehner, F.M.E; Naber, K.G. Urinary Tract Infections – General. Antimicrobe.

L82 there are regulations for microbial taxonomy writing. Please find them at the MDPI guidelines for authors. Please fix the name of the microorganism.

Response:

In order to satisfy your request, we have modified the text, curent line 81.

L84-L87 The taxonomy must be fixed. Please make sure that you are using the correct term of Enterobacteriacae or Enterobacteriales. The statement needs reference.

Response:

In order to satisfy your request, we have corrected text: current lines 83, 84. We have also added the following reference:

Flores-Mireles, A.L.; Walker, J.N.; Caparon, M.; Hultgren, S.J. Urinary tract infections: epidemiology, mechanisms of infection and treatment options. Nat Rev Microbiol. 2015, 13(5): 269–284.

L87 the reference No 47 (Echols, R.M.; Tosiello, R.L.; Haverstock, D.C.; Tice, A.D. DemograpHic, clinical, and treatment parameters 493 influencing the outcome of acute cystitis. Clin Infect Dis 1999, 29(1), 113.) is not providing data supporting the statement (In athletes with recent 86 hospitalizations, bacterial such as Pseudomonas, Enterococci and Staphylococci have also 87 been found [47].)

Moreover the taxonomic writing of the organisms must be fixed.

Response:

In order to satisfy your request, we have corrected the text: current lines . We have also modified the reference with the following one:

Jaworski, C.A. Valerie Rygiel, V. Acute illness in the athlete. Clin Sports Med. 2019 Oct; 38(4): 577–595.

L87-L93 the part requires attention and re-writing/clarification. The part is missing reference indicating the source of the information.

Response:

In order to satisfy your request, we have modified the text: curent line from 85 to 92. Moreover, we have added the following references:

Oresnstein, R.; Wong, E.W. Urinary tract infections in Adults. Am Fam Physician. 1999 Mar 1;59(5):1225-1234.

L93-95 Unfortunately, 20% of 94 people experience a second infection after the first (recurrent infections), and some people suffer from 95 it for a long time and incessantly (recurrent infections).

It is unclear where authors find the results. The reference must be included.

Response:

In order to satisfy your request, we have modified the text: current line 93. We have also added the following reference:

Lindsay, E.N. Prophylaxis: recurrent urinary tract infection in women. Infection. 1992, 20: S203-S205.

L95 Sometimes, the bacterial strains in cases of recurrence are different from those that have infected the first time (acute infection), on the other 97 hand, uropathogens invade the structures of the urinary tract, becoming a chronic presence, 98 therefore, they find themselves protected from drug therapies.

Why sometimes the strain is different while other times the strain is the same? The reference is missing.

Response:

There are two possible pathways in the pathogenesis of recurrent UTI; frequent repeat ascending infections and chronic/persistent infections in the bladder. Each pathway also might result from two possible mechanisms: bacterial factors and deficiencies in host defense.

(Jia-Fong Jhang and Hann-Chorng Kuo. Recent advances in recurrent urinary tract infection from pathogenesis and biomarkers to prevention. Ci Ji Yi Xue Za Zhi. 2017; 29(3): 131–137).

We have also added this reference in the text.

L98-L99 Therefore, a urine screening of athletes 99 can help identify the infection, the bacterium and for an immediate targeted antibiotic treatment.

Why in this case athletes are selected population with increased demand to be screened for recurrent UTI’s?

Response:

Through examination of the urine, variations in the bacterial, leukocyte and other dosed analytes are easily identifiable. These changes act as alarm bells and allow you to intervene promptly in order to safeguard the athlete's health and avoid performance drops.

L105 requires attention.

Response:

In order to satisfy your request, we have modified the text: current line 107.

The overall introduction requires great attention.

L140 requires attention.

Response:

In order to satisfy your request, we have modified the text: line 140.

2.3 Are those values in between the brackets indicating the diagnostic range or normal values?

Response:

The values ​​shown in the paragraph 2.3 correspond to the reference values ​​used in our laboratory.

L151 proteins (0 e 20 mg/dl) should e be replaced by -?

Response:

In order to satisfy your request, we have modified the text: current line 156.

L162 authors did not provided the rationale when the study design required to use a microscopy.

Response:

In presence of fungal spores or red blood cells values unusually high compared to hemoglobin values, examination via optic microscope was conducted, due to red blood cells and spores being possibly misinterpreted by the cytometer.

L166 p should be italic.

Response:

In order to satisfy your request, we have modified the text: now line 169

Figure 1. Again, I would recommend to simplify the presentation of the data by changing the style of the graphs to column graphs. Di authors used Grappad to prepare such graphs as indicated by the methods part?

The units presented in the pie graphs are unknown. It is not clear what units were used for the graph construction. The description indicates that authors used mean ±SD, although I am not sure how mean values were calculated. Please provide the clear statement.

Response:

In order to satisfy your request, we have simplified both figure 1, in fact we have reported the graphs for the pH and for the specific weight. The values ​​shown are the average of the values ​​obtained by the individual 12 players +/- the SD. We then performed ANOVA as a statistical analysis.

L203 authors did not monitor bacterial populations. Please fix the statement, since the used methodologies are not allowing to monitor the bacterial populations.

Response:

We monitored the presence of bacterial via optic microscope.

L206 the same problem regarding bacterial populations.

Response:

We monitored the presence of bacterial via optic microscope.

L220—L221 Furthermore, from a microscopic analysis, of the samples containing bacteria we have 221 highlighted the presence of Escherichia Coli.

The taxonomy is wrong. Moreover and the most important, the microscopy will NEVER allow one to identify the species. All that is possible (without staining) is to differentiate the morphology. The statement is very wrong and not correct in many aspects.

Response:

It is known that a few drops of urine analyzed under a microscope can identify the presence of E coli; in addition we performed urine culture to confirm the data.

We have therefore modified the text to be more precise.

Fig 2. Fig 3 again the units are not clear. Same problem as Fig 1.

Response:

We changed the figure and added a table. The figure shows the unit of measurement used.

Discussion

The manuscript lacks of discussion of the results. The authors do not fully compares their results with similar results observed by other authors.

It is still unclear why authors decided to analyze male subjects and not females or both gender. There is no provided rationale for the subject selection in the discussion.

The manuscript contains many typos.

Response:

In order to satisfy your remarks, we have changed some of the discussion considerations.

as far as our study is concerned, we monitored 12 male athletes as we analyzed only one male team.

It will be our interest to expand our studies and also compare them with females.

We have corrected and edited English to improve the manuscript

Reviewer 3 Report

The manuscript titled:"Urinary biomarkers: diagnostic tool to monitor athletes' health status" from Pero et al. describes a method based on monitoring biochemical parameters present in urine during competitive physical activity to prevent the onset and hence timely treatment of common urinary infections in athletes. The authors aim at identifying urinary biomarkers by evaluating several parameters associated with the athlete's health status in a group of 12 men professional basketball players. This is the first report describing all of these parameters in a population of competitive athletes before, during and after strenuous physical activity. The authors observe changes at the 1 months timepoint that are in line with fatigue, hypohydration and the possible appearance of infections supporting the usefulness of a simple, non-invasive test that allows monitoring the athlete's health for a timely intervention. Overall, this manuscript is of interest and adds knowledge to the specific field for which there are still few recommendations for biomarker panels for tracking changes in individuals participating in competitive physical activity.

The reviewer recommends the manuscript to be accepted after the following minor revisions:

  • Extensive editing of the English language is highly recommended.
  • Punctuation and spelling should also be revisited, not only in the text but also in figure legends.
  • Statistical analysis: the authors are comparing 3 measurement from the same individuals, over time. It is recommended to analyze statistical differences by the “repeated measures ANOVA” and to change the results accordingly.
  • Participants: 15 players are mentioned on line 136 but 12 are reported throughout the manuscript. Please, reconcile.
  • Line 45: the authors mention the role of the microbiome (assumingly gut microbiome) in athlete’s health. It would be helpful if they could expand a little on this concept and link the microbiome to UTIs.
  • Line 158 (Biochemical determinants): please, list the parameters tested. Also, the purpose of microscope examination.
  • Results: there is a high amount of numbers (percentages) in the text. For an easier reading, maybe the authors would want to consider to include those numbers in the figures (in the pie charts or in brackets next to the units) or put them in a table, as they did in Table 1.
  • Results: at the end of each (or groups of) examined parameter, authors comment the results by stating that there is “a decrease overtime”. Because in the conclusions the authors state that the changes associated with possible pathological features are observed between 0 and 1 month, it is suggested to revisit the results commenting on the alterations detected at 1 month, as well.
  • 1A,B,C: please, change ph to pH
  • 2D: space is missing between the letter D and the text. Moreover, in this figure are shown the wrong units for Squamous Cells. Please, check on that.
  • Conclusions: on line 305 the authors describe a “decrease in the SW over time” . On the graph in Fig. 1P an increase is shown (Fig. 1P) and on line 318 the authors discuss an increase in SW. Please, reconcile.

Author Response

Comments and Suggestions for Authors

The manuscript titled:"Urinary biomarkers: diagnostic tool to monitor athletes' health status" from Pero et al. describes a method based on monitoring biochemical parameters present in urine during competitive physical activity to prevent the onset and hence timely treatment of common urinary infections in athletes. The authors aim at identifying urinary biomarkers by evaluating several parameters associated with the athlete's health status in a group of 12 men professional basketball players. This is the first report describing all of these parameters in a population of competitive athletes before, during and after strenuous physical activity. The authors observe changes at the 1 months timepoint that are in line with fatigue, hypohydration and the possible appearance of infections supporting the usefulness of a simple, non-invasive test that allows monitoring the athlete's health for a timely intervention. Overall, this manuscript is of interest and adds knowledge to the specific field for which there are still few recommendations for biomarker panels for tracking changes in individuals participating in competitive physical activity.

The reviewer recommends the manuscript to be accepted after the following minor revisions:

Extensive editing of the English language is highly recommended.

Response

Thanks to the Reviewer 1 for the helpful comments.

Punctuation and spelling should also be revisited, not only in the text but also in figure legends.

Response:

In order to satisfy your request, we have modified the text: current lines: 136, 184, 199, 238, 330, 367.

Statistical analysis: the authors are comparing 3 measurement from the same individuals, over time. It is recommended to analyze statistical differences by the “repeated measures ANOVA” and to change the results accordingly.

Response:

To improve the statistical analysis we carried out the ANOVAs followed by Bonferroni or Tukey's multiple comparison tests

Participants: 15 players are mentioned on line 136 but 12 are reported throughout the manuscript. Please, reconcile.

Response:

In order to satisfy your request, we have modified the text: current lines 28, 140, 181.

Line 45: the authors mention the role of the microbiome (assumingly gut microbiome) in athlete’s health. It would be helpful if they could expand a little on this concept and link the microbiome to UTIs.

Response:

In order to satisfy your request, we have modified the text: current lines from 72 to 75.

Neugent, M.L.; Hulyalkar, N.V.; Nguyen, V.H.;  Zimmern, P.E.; De Nisco, N.J Advances in Understanding the Human Urinary Microbiome and Its Potential Role in Urinary Tract Infection. American Society for Microbiology. 2020, 11(2), 1-15.

Line 158 (Biochemical determinants): please, list the parameters tested. Also, the purpose of microscope examination.

Response:

The list of parameters tested has been reported in paragraph 2.3 from current lines 154 to 157.

Results: there is a high amount of numbers (percentages) in the text. For an easier reading, maybe the authors would want to consider to include those numbers in the figures (in the pie charts or in brackets next to the units) or put them in a table, as they did in Table 1.

Response:

In order to satisfy your request, we have created the tables in the text.

Results: at the end of each (or groups of) examined parameter, authors comment the results by stating that there is “a decrease overtime”. Because in the conclusions the authors state that the changes associated with possible pathological features are observed between 0 and 1 month, it is suggested to revisit the results commenting on the alterations detected at 1 month, as well.

Response:

To satisfy your requests, we have reviewed the results.

1A,B,C: please, change ph to pH

Response:

In order to satisfy your request, we have corrected the Figure 1.

2D: space is missing between the letter D and the text. Moreover, in this figure are shown the wrong units for Squamous Cells. Please, check on that.

Response:

In order to satisfy your request, we have corrected the Figure 2

Conclusions: on line 305 the authors describe a “decrease in the SW over time” . On the graph in Fig. 1P an increase is shown (Fig. 1P) and on line 318 the authors discuss an increase in SW. Please, reconcile.

Response:

We modified the text

L44 requires attention. Perhaps there were some formatting problems ( co μL d)

Response:

In order to satisfy your request, we have modified text: now lines 42, 49, 114, 120, 128, 192, 216, 219, 221, 222, 224, 225, 227, 228, 229, 230, 232, 262, 265, 289, 330, 333, 334, 351, 362, 363, 364, 401, 402, 410, 420, 428, 429, 431, 437, 440, 432, 435, 473, 474, 439, 512, 529, 539, 547, 555.

We then performed the English editing to improve the manuscript.

The changes made are highlighted in yellow in the text

Round 2

Reviewer 2 Report

Dear Pero et al.

The manuscript now is greatly improved. The authors did address all of the comments/suggestions indicated during the previous review. The overall look of the manuscript is good.

I have a few minor comments/suggestions for the authors to address.

A very minor spell check is needed. The minor check on symbols is needed as well. The overall quality of language is good.

L290-L231 Authors state: Furthermore, the microscopic analysis showed that Escherichia coli was present in all of the 229samples containing bacteria.In addition to confirm the presence of the bacterium, urine culture was 230performed.

In the previous review, the question was raised regarding the possibility to identify the genus of bacteria with microscopy. I do not agree with the following statement and the provided answer.

The identification of Escherichia coli by microscopy is not possible unless FISH, immunofluorescence or other diagnostic methodology is used. By using microscopy authors were able to state that rod shape bacteria are present. It is also possible to make a comment on the motility but for sure not on the genus. 

For example, Klebsiella and E. coli microscopically in urine would look the same. Proteus, Pseudomonas, or any motile rod shape organism of similar size would look exactly the same as E. coli.

Before proceeding with publishing the manuscript, the mentioned part must be fixed.

best,

P.

Author Response

Dear Editor, 

thank you for the helpful comments about our manuscript entitled “Urinary Biomarkers: diagnostic tool to monitor athletes’ health status”, submitted to International Journal of Environmental Research and Public Health (IJERPH). We have appreciated the comments received and have carefully re-considered them in preparing a new version of the manuscript.

A point-by-point response to the comments is attached below.

We believe that the manuscript is now significantly improved thanks to your suggestions.

We hope that the new version of the paper deserve publication on the International Journal of Environmental Research and Public Health.

Best regards,

Prof.Dr Olga Scudiero and Prof. Dr. Raffaela Pero

Reviewer 2

Comments and Suggestions for Authors

Dear Pero et al.

The manuscript now is greatly improved. The authors did address all of the comments/suggestions indicated during the previous review. The overall look of the manuscript is good.

I have a few minor comments/suggestions for the authors to address.

A very minor spell check is needed. The minor check on symbols is needed as well. The overall quality of language is good.

L290-L231 Authors state: Furthermore, the microscopic analysis showed that Escherichia coli was present in all of the 229samples containing bacteria.In addition to confirm the presence of the bacterium, urine culture was 230performed.

In the previous review, the question was raised regarding the possibility to identify the genus of bacteria with microscopy. I do not agree with the following statement and the provided answer.

The identification of Escherichia coli by microscopy is not possible unless FISH, immunofluorescence or other diagnostic methodology is used. By using microscopy authors were able to state that rod shape bacteria are present. It is also possible to make a comment on the motility but for sure not on the genus.

For example, Klebsiella and E. coli microscopically in urine would look the same. Proteus, Pseudomonas, or any motile rod shape organism of similar size would look exactly the same as E. coli.

Before proceeding with publishing the manuscript, the mentioned part must be fixed.

Response:

In order to satisfy your request, we have modified lines 229 and 230, describing in an appropriate way the steps taken for the identification of the E. coli bacterium. Finally, to specify the diagnostic procedure we have added references to support the analysis conducted.

The changes made are highlighted in green in the text
